# Integrative Thermodynamic Strategies in Microbial Metabolism

**DOI:** 10.3390/ijms262210921

**Published:** 2025-11-11

**Authors:** Martijn Bekker, Oliver Ebenhöh

**Affiliations:** 1Data Science & Informatics and Supply Chain Design, Wageningen University and Research, 6708 WG Wageningen, The Netherlands; 2Institute of Quantitative and Theoretical Biology, Department of Biology, Heinrich-Heine University Düsseldorf, 40225 Düsseldorf, Germany

**Keywords:** thermodynamics, metabolism, microorganisms, biotechnology, catabolism

## Abstract

Microbial metabolism is intricately governed by thermodynamic constraints that dictate energetic efficiency, growth dynamics, and metabolic pathway selection. Previous research has primarily examined these principles under carbon-limited conditions, demonstrating how microbes optimize their proteomic resources to balance metabolic efficiency and growth rates. This study extends this thermodynamic framework to explore microbial metabolism under various non-carbon nutrient limitations (e.g., nitrogen, phosphorus, sulfur). By integrating literature data from a range of species, it is shown that growth under anabolic nutrient limitations consistently yields more negative Gibbs free energy (ΔG) values for the net catabolic reaction (NCR) per unit of biomass than carbon-limited scenarios. The findings suggest three potentially complementary hypotheses: (1) proteome allocation hypothesis: microbes favor faster enzymes to reduce the proteome fraction used for catabolism, thus freeing proteome resources for additional nutrient transporters; (2) coupled transport contribution hypothesis: the more negative ΔG of the NCR may in part stem from the increased reliance on ATP-coupled or energetically driven transport mechanisms for nutrient uptake under limitation; (3) bioenergetic efficiency hypothesis: microbes prefer pathways with a more negative ΔG to enhance the cellular energy status, such as membrane potentials or the ATP/ADP ratio, to support nutrient uptake under anabolic limitations. This integrative thermodynamic analysis broadens the understanding of microbial adaptation strategies and offers valuable insights for biotechnological applications in metabolic engineering and microbial process optimization.

## 1. Introduction

Microbial metabolism is governed by the fundamental principles of thermodynamics, constraining energy efficiency, growth rates, and metabolic trade-offs [1,2,3,4,5,6,7]. Understanding the Gibbs free energy (ΔG) requirements of different metabolic pathways is critical for predicting microbial behavior under various environmental constraints.

The mosaic approach framework of non-equilibrium thermodynamics, described by Westerhoff et al. [1,2,8], provides a theoretical basis for understanding microbial growth and the associated thermodynamic forces involved. This approach views microbial metabolism in terms of energy fluxes, distinguishing between the output flow (catabolic processes) and input flow (substrate utilization). A key insight from this perspective is that microbial systems are often, though not universally, optimized toward the maximum growth rate rather than yield, depending on the nutrient availability and environmental stressors [1,6,9,10]. Interestingly, under nutrient limitations, deviations from theoretical ATP yields ($Y_{\mathrm{ATP}}$) arise due to maintenance processes, which include balancing ion leakage and repairing oxidative damage, futile cycling [11,12,13], and variation in the coupling efficiencies of proton translocators, leading to altered P/O ratios. According to Westerhoff [8], these deviations emphasized the need to account for non-growth-related maintenance requirements.

Flamholz et al. [9] approached thermodynamic modeling of microbial metabolism by use of the Gibbs free energy equation in combination with the Haldane rate law, which defines the limitations (with respect to the forward and reverse $k_{\mathrm{cat}}$ and $K_{M}$) of enzyme characteristics. Using this aspect Flamholz et al. applied the rules for a single enzymatic reaction (i.e., the Haldane equation) to complete metabolic pathways [9]. This allowed for an explanation of the presence of the Entner–Doudoroff (ED) pathway and its preferential use in some cases over the Embden–Meyerhoff–Parnass (EMP) pathway, which appears counter-intuitive considering that the ED pathway only produces one ATP per glucose converted to pyruvate in comparison with the two ATPs of the EMP pathway. It was eloquently shown that to obtain the same ATP production rate, the proteome resource requirements for the ED pathway would be 5-fold lower (using measured kinetics) than for the EMP pathway. Under the assumption of comparable energy charges, by producing only one ATP instead of two, the net ΔG of the ED pathway is more negative than that of the EMP pathway [9]. This, in turn, allows for higher $k_{\mathrm{cat}}$ values of the forward reactions allowed within the limitations of the more negative ΔG of the ED pathway as compared with the EMP pathway [9,14], leading to faster ATP production rates with a smaller enzyme investment, despite the twofold lower yield of ATP per glucose converted to pyruvate.

This hypothesis was subsequently expanded to explain overflow metabolism in *Escherichia coli* [6,14,15,16,17], the Crabtree effect [18,19,20], and mixed acid versus homolactic fermentation [21]. By using these concepts it can be hypothesized that all these observations are a consequence of proteome resource allocation constraints [3] imposed by ΔG and ATP yield trade-offs [22]. These are of high relevance for a better understanding of the intricacies with which thermodynamics influence the selected metabolic pathways for ATP generation and the consequences of this for proteome resource allocation.

The above endeavors all focus on explaining the change from carbon-limited growth conditions to batch conditions (i.e., carbon-excess conditions) [6,21,23]. None of these studies have focused on non-carbon-limited growth conditions. Interestingly, such conditions show a much higher maintenance requirement than carbon-limited growth [13,24,25,26] and calculations of the qATP_m_ indicated higher maintenance, especially at low growth rates [24]. However, these observations are not yet included in recent studies on thermodynamics of microbial growth.

The motivation behind this research study is therefore to extend previous studies with an integrative thermodynamic analysis of published literature of non-carbon limitations and the effects on the thermodynamics of growth. We systematically compare the energetics of microbial growth in a large number of chemostat experiments encompassing different microbial species, nutrient limitations, and growth conditions. For this, we introduce the concept of the net catabolic reaction, which signifies the catabolic conversion of nutrients that is associated with the formation of a defined amount of biomass. Because the separation of metabolism into anabolism and catabolism is not trivial and in parts also ambiguous, our definition is to some degree arbitrary. However, for the goal of providing a systematic comparison, the precise choice of definition is secondary as long as a clear and consistent definition is provided that can be applied to all investigated scenarios. To our knowledge, our approach employing the net catabolic reaction presented here is the first attempt to provide a clearly defined, widely applicable quantification of the energetic costs of biomass formation, allowing for systematic comparisons between different organisms and nutrient limitation regimes. By utilizing this extended theoretical model, we aim to widen our understanding of the fundamental thermodynamic phenomena observed under specific nutrient limited conditions [24,25,27].

Microbial growth is often conceptually viewed as a thermodynamic energy converter, in which the energetically favorable reactions of catabolism drive anabolism [1,28,29]. The ‘coupling agent’ is the energy currency metabolite ATP, which is produced in catabolic pathways, exploiting the free energy gradient of the conversion of nutrients into catabolic products. This ATP, in turn, drives thermodynamically unfavorable reactions of anabolism (see Figure 1 for a schematic representation). While ATP is a key intermediate and quantities such as $Y_{\mathrm{ATP}}$, i.e., the ATP yield per mole substrate consumed, are important characteristics for the thermodynamic efficiency of growth, these are notoriously difficult to measure and are often inferred using metabolic models and rely on certain assumptions, such as the P/O ratio. We here propose an approach that does not depend on the knowledge of ATP yields or demands, but that instead relies exclusively on knowledge of the overall macrochemical growth equation.

An important quantity in the center of our focus is the amount of Gibbs free energy that needs to be released by catabolic processes to produce one carbon-mole of biomass. By measuring the catabolic energy requirement to produce one carbon-mole of biomass, this quantity provides an important thermodynamic characteristic of microbial growth. We will in the following refer to it as the catabolic Gibbs free energy and denote it with $\Delta G_{X/S}$.

The quantity $\Delta G_{X/S}$ is determined from the macrochemical growth equation as follows: First, the macrochemical growth equation is normalized to the production of one carbon-mole of biomass. Second, the net anabolic reaction to form one carbon-mole of biomass is subtracted, leaving only the catabolic conversion involved in the formation of one carbon-mole of biomass. We term the remaining chemical conversion the net catabolic reaction (NCR). In order to define the net anabolic reaction, we consider an idealized process that converts nutrient carbons into biomass with a maximally possible yield. If the biomass is less reduced than the substrate, the theoretical carbon yield of this anabolic process is 1 (100%), assuming the presence of external electron acceptors, such as oxygen. If the biomass is more reduced than the substrate, some nutrient carbons need to be oxidized to maintain the redox balance, with a maximal theoretical carbon yield of $\frac{\gamma_{S}}{\gamma_{X}}$, where $\gamma_{S}$ and $\gamma_{X}$ represent the degrees of reduction of the substrate and biomass, respectively. The NCR can easily be determined from a macrochemical growth equation (see [Fig ijms-26-10921-i001]) and we calculate the associated reaction energy $\Delta G_{X/S}$ using the eQuilibrator API [9].

**Figure ijms-26-10921-i001:** 

It should be noted that the ‘ideal’ anabolic reaction is an abstract concept and there is no guarantee that such a reaction can actually be realized. Interestingly, though, we recently showed with genome-scale metabolic models that feasible flux distributions indeed do exist and closely approximate (>95%) the theoretical carbon yield, and they are fully operable provided sufficient ATP is provided by other pathways [30]. However, whether such idealized pathways actually exist is secondary to our investigation. Instead, a simple, unambiguous definition is important since it gives rise to a single calculation method that can be applied to all micro-organisms, carbon sources, and conditions. The NCR and the associated catabolic free energy $\Delta G_{X/S}$ fulfill all these conditions, and thus, they provide a thermodynamic measure of microbial growth that is comparable across organisms and growth conditions.

## 2. Results

To broaden our thermodynamic understanding of microbial growth to non-carbonlimited conditions, we surveyed the literature reporting both carbon- and non-carbon-limited growth under comparable experimental conditions. We limited our search to articles where carbon-limited or anabolic-limited continuous chemostat conditions were used and gas exchange, nutrient consumption, and production formation rates were systematically measured to allow for reconstruction of a balanced growth equation. In total, we selected 10 articles that fulfilled these criteria by studying various yeasts, including *Saccharomyces cerevisiae*, *Cyberlindnera jadinii*, and *Pichia kluyveri* [18,19,25,31,32,33,34]; *E. coli* [35]; *Klebsiella aerogenes* [36]; and *Klebsiella pneumoniae* [11,31]. In addition, we included one study on *E. coli*, in which gas exchange was not measured, to also include iron limitation in our analysis [37,38]. To obtain the thermodynamic characteristics, for each experiment, we first determined a balanced macrochemical growth equation that best fit the experimentally determined exchange fluxes of nutrients and metabolic products and the set dilution (growth) rate. This fitting was necessary because the carbon recovery coefficients diverged from 100%. This equation was normalized to the production of 1 C-mol biomass, and the thermodynamic characteristics were calculated as described in Section 4 and [Fig ijms-26-10921-i001]. Our strategy extends previous approaches [10,30] by providing an estimate of the thermodynamics of catabolism normalized per unit of biomass formed and including non-carbon or non-energy limitations in our research [8,30].

We started our analysis with the study of Tai et al. [25] because it contains a large dataset of *S. cerevisiae* under both aerobic and anaerobic conditions and four nutrient limitations (carbon, nitrogen, phosphorus, and sulfur). We observed that the $\Delta G_{X/S}$ is more negative under all non-C-limited conditions (i.e., nitrogen, phosphorus, and sulfur limitations) than during C-limited growth (see Table 1) under both aerobic and anaerobic conditions. The data demonstrate that under nutrient stress (i.e., nitrogen, phosphorus, or sulfur limitation), the yeast cells require significantly more catabolic energy to sustain biomass production both under aerobic and anaerobic conditions. Our results indicate a clear pattern of increased catabolic energy demand with nutrient limitations.

Second, we analyzed the available *K. pneumoniae* data (see Table 2 and [12]), which included chemostat experiments for two different nitrogen sources (ammonia and nitrate) under glucose-, nitrogen-, phosphorus-, and potassium-limited conditions. Again, determining the catabolic Gibbs free energy $\Delta G_{X/S}$, we observe that under nitrogen, phosphorus, and potassium limitations, these values were consistently more negative than under carbon limitation.

To confirm these initial observations, we also included the other relevant articles that studied microbial growth under various nutrient limitations. Again, we applied the same strategy as above and determined the catabolic Gibbs free energy $\Delta G_{X/S}$ for every single reported chemostat experiment. This allowed us to use a broader dataset encompassing multiple microbial species, a variety of nutrient limitations (carbon, sulfate, nitrogen, phosphate, potassium, iron), and diverse carbon sources (glucose, ethanol, glycerol, mannitol, lactate, ketoglutarate). The entirety of the analyzed data is summarized in Figure 2.

Interestingly, first focusing on carbon limited conditions only (circles in Figure 2) reveals that despite variations in the microbial species, media composition (all chemically defined), and dilution rates, the catabolic Gibbs free energy $\Delta G_{X/S}$ appears rather independent of the organism, growth rate, and type of catabolic carbon source used. It is noteworthy that both Crabtree-positive (light green circles) and Crabtree-negative (dark green circles) yeast strains show a similar $\Delta G_{X/S}$ at all dilution rates despite the drastically different catabolic routes and the concomitantly different Gibbs free energy yield per carbon-mole of substrate. Further partitioning $\Delta G_{X/S}$ into contributions from respiratory and non-respiratory catabolism revealed that the respiratory component declines sharply with increasing growth rate, yet the total $\Delta G_{X/S}$ remains rather constant (see Figure 3). Notably, carbon-limited growth on ethanol seems to result in a slightly more negative $\Delta G_{X/S}$ of the NCR (see orange circles in Figure 2 and [33]).

In contrast, under anabolic nutrient limitations (e.g., sulfur, nitrogen, phosphorus), the $\Delta G_{X/S}$ of the NCR is consistently more negative than for carbon limited conditions. This pattern holds true across all examined organisms and conditions, indicating that anabolic limitations may universally impose a greater thermodynamic demand on catabolism than carbon limitation.

## 3. Discussion

Our analysis of nutrient-limited microbial growth provides additional insights on top of the existing literature that discusses the trade-offs between proteomic investment and thermodynamic efficiency under various growth rates under continuous C-limited growth conditions. There, it was consistently observed that under high growth rates, catabolic fermentation products are excreted. This entails that the energetic substrate use efficiency decreases because less ΔG is obtained per C-mol of substrate while the yield simultaneously decreases, and more substrate is consumed per C-mol of biomass produced. This behavior is commonly explained [6,9,14] by arguing that the less efficient catabolic routes require fewer proteins but display a higher rate ($k_{\mathrm{cat}}$) and that, therefore, the same ATP production rate can be achieved with a smaller proteome fraction, leaving more space for anabolic proteins, in particular the ribosome. We think this hypothesis still holds, yet we cannot apply it to non-carbon limited conditions given that (1) under all non-C-limited growth conditions, the $\Delta G_{X/S}$ of the overall growth equation is significantly more negative than under C-limited growth conditions for all microorganisms studied at similar growth rates; (2) the $\Delta G_{X/S}$ of the NCR is similar under both aerobic and anaerobic conditions (see the dark green circles in Figure 2 and Table 1); (3) the effect is observed independent of the growth rate or carbon limitation used in the experimental setup. Moreover, it should be noted that knowledge of the catabolic equation alone does not allow for discriminating between different catabolic pathways, and therefore, also does not provide quantitative insight into the actual qATP. For example, both the ED and EMP pathways may result in metabolizing one mole of glucose into two moles of pyruvate, while in the former case, one mole of ATP was produced and in the latter two. Using the approach described here omits the use of any assumptions of catabolic pathways used and ATP produced and thereby circumvents this challenge. In fact, if we consider two organisms, one of which uses the ED and the other the EMP pathway, but which are otherwise identical and grow at identical rates, then the ED-using organism would need a catabolic flux that was twice as fast as the EMP-using organism to provide the same amount of ATP per unit time. With our approach, we observed that the ED-using organism requires more free energy from catabolism to produce 1 C-mol of biomass. However, whether this apparent increased energy demand is actually caused by the usage of a lower-yield pathway or whether other energetic demands (such as those caused by nutrient scarcity or external stresses) are present cannot be discriminated from knowledge of the net catabolic reaction alone.

A remarkable observation of our analysis is that for C-limited conditions, the catabolic Gibbs free energy $\Delta G_{X/S}$ per C-mol of biomass produced remains rather constant, with values around $\Delta G_{X/S}=300\ldots 400\;\mathrm{kJ}\;{\mathrm{mol}}^{-1}$ (all circles in Figure 2). This value is relatively constant, irrespective of the organism investigated (*S. cerevisiae* in Rieger et al. [18] and van Hoek et al. [19], *E. coli* in Kayser et al. [35] and Folsom and Carlson [38], and *K. pneumoniae* in Buurman et al. [12] and Simons et al. [31]), the carbon source used, or the dilution rate applied. Most interestingly, this value is independent of whether ethanol (or any other overflow product) is released or not (compare the light and dark green circles in Figure 2 for data on the Crabtree-positive yeast *P. kluyveri* and the Crabtree-negative yeast *S. cerevisiae* and see Figure 3 for three classical overflow experiments). This clearly indicates that under C-limited conditions, despite a rather drastic shift in metabolic pathways, the overall energetic requirement to produce 1 C-mol of biomass is rather independent of the growth rate. A slight increase in $\Delta G_{X/S}$ is visible for the yeast strains grown on ethanol (yellow points in Figure 2, [33]) at low growth rates (≲$0.1\;h^{-1}$). However, for faster growth rates, the $\Delta G_{X/S}$ value is no longer increased. While proteome allocation principles may indeed explain the shift from respiratory to fermentative pathways, it does not make a statement regarding the overall catabolic energy requirement for growth.

To understand the above phenomena where the net catabolic Gibbs free energy $\Delta G_{X/S}$ is systematically more negative for limitations other than carbon, we think it relevant to highlight that during nutrient limitation, importing the limiting nutrient is expected to result in a more positive ΔG compared with conditions where nutrients are abundant, as the concentration gradient from extracellular to intracellular space is higher during nutrient limitations. This more positive ΔG of nutrient uptake will add to the overall positive ΔG of biomass formation, where disordered nutrients are concentrated and combined into a structured living cell [28]. This effect is not included in our calculations given that no information is available on the extracellular concentrations of the limiting nutrients.

The origin of the more negative $\Delta G_{X/S}$ of the NCR under anabolic-nutrient-limited conditions remains unclear. This can either be caused by energy dissipation reactions or by overall changes in the catabolic pathways used, and possibly both. If one understands a growing microbe as an energy converter (see, e.g., [29]), in which the negative ΔG of catabolism drives the (slightly positive) ΔG of anabolism, increasing the negative catabolic ΔG in non-carbon-limited conditions makes intuitive sense. Nutrient limitations increase the energetic requirements to obtain these nutrients, e.g., by using active transporters or expressing extracellular pathways to capture sparse nutrients. Hence, an increased driving force of catabolism seems necessary to overcome the increased energy barrier of the driven anabolism. In the following, we postulate three, non-mutually exclusive hypotheses for the mechanisms underlying a competitive advantage for the more negative ΔG under anabolic nutrient limitations:
Proteome allocation hypothesis: The more negative $\Delta G_{X/S}$ of the NCR under nutrient-limited conditions is a result of the preferential use of catabolic pathways with a lower ATP yield per substrate, and thus, a more negative $\Delta G_{X/S}$ (normalized to a carbon-mole of new biomass formed). These pathways require fewer proteomic resources for ATP generation [3,14,16], freeing up a larger fraction of the proteome for the expression of nutrient transport systems and ribosomes. This redistribution could confer a selective advantage by enhancing the uptake of the limiting nutrient.Coupled transport contribution hypothesis: The more negative $\Delta G_{X/S}$ of the NCR may in part stem from the increased reliance on ATP-coupled or energetically driven transport mechanisms for nutrient uptake under limitation. When nutrients are scarce, cells may increasingly use active transporters that directly couple ATP hydrolysis to substrate import. While this incurs an energetic cost, it renders the overall nutrient uptake process thermodynamically more favorable and may contribute to the net negative ΔG of catabolism required to support growth. This strategy could enhance the substrate import efficiency under nutrient limitation, thereby offering a selective advantage despite the higher energetic investment.Bioenergetic efficiency hypothesis: The use of catabolic pathways with a more negative ΔG may also lead to increased cellular energy states, such as higher ATP/ADP ratios or higher membrane potentials. We hypothesize that this bioenergetic enhancement could improve the functionality of transporters that rely on ATP hydrolysis or membrane potential, thus supporting faster nutrient uptake under the anabolic limiting conditions discussed in this perspective. This could help overcome the additional thermodynamic burden of transporting the low-concentration extracellular limiting nutrient to the highly concentrated intracellular environment of this nutrient.

All three strategies may contribute to the observed changes in the overall $\Delta G_{X/S}$ of the NCR and it will prove scientifically challenging to disprove one or the other. In fact, all of the proposed mechanisms will result in a more negative catabolic Gibbs free energy $\Delta G_{X/S}$, and may therefore contribute to energetically and kinetically overcoming nutrient limitations. Which of these mechanisms are actually active can be studied by detailed pathway analysis using stable isotope ^13^C labeling experiments that allow for discrimination of the specific pathways used during specific conditions [39,40]. In addition, experimental measurements of the energy status and the membrane potential across the mitochondrial inner membrane would allow for testing the bioenergetic efficiency hypothesis. This may still however be challenging given that some of the microorganisms analyzed in this study show a branched respiratory chain that includes respiratory enzymes that differ in the number of protons translocated per oxygen reduced. Examples are the NDHII complex and cytochrome bd-I, which both show a lower proton translocation than NDHI and cytochrome bo, yet the $k_{\mathrm{cat}}$ values of these enzymes are significantly higher than their high-efficiency counterparts [15,41,42,43]. It would therefore be relevant to further explore the catabolic routes and proteome resource allocation dynamics under anabolic nutrient limitations to provide further insights into the potential cause of the changes in ΔG of the NCR. Indications can be found in the literature that this indeed may play a role for the cytochrome bd-II complex in *E. coli*, which shows increased expression under phosphate-limited conditions [26,44], implying that under phosphate-limited continuous growth, this catabolic pathway may play a role in the changes in the overall $\Delta G_{X/S}$.

In general it is observed that although there is a strong shift in the catabolic pathways used [6,18,21,35], the overall ΔG of the NCR does not show a significant change (see Figure 2) when switching from C-limited growth conditions to non-C-limited growth conditions (i.e., batch growth). Although the ΔG per calculated ATP produced may change when such pathways are used (see Table 3 for some standard example calculations), it should be noted that such calculations assume specific catabolic pathways that may practically vary given the flexibility microorganisms tend to have in the respiratory chain (and therefore, the P/O ratio of respiration) and in the used phosphorylation pathways. Therefore ATP-based calculations are very challenging and we therefore suggest using the NCR to ensure a correct interpretation of the thermodynamics of catabolism.

## 4. Materials and Methods

### 4.1. Net Catabolic Reaction

Every dataset is individual. We therefore first converted every single chemostat dataset so that all measured exchange fluxes were given in the unit $C-\mathrm{mol}\;{\left(C-\mathrm{mol}\;\mathrm{biomass}\right)}^{-1}$ or $\mathrm{mol}\;{\left(C-\mathrm{mol}\;\mathrm{biomass}\right)}^{-1}\;h^{-1}$ for non-carbon-containing compounds (in particular O_2_). These values were assembled in a vector so that consumed compounds were counted as negative. For convenience, we further assigned a biomass to the index 0. For *n* additionally measured exchange fluxes, this resulted in an n+1-dimensional vector $e={\left(e_{0},\ldots ,e_{n}\right)}^{T}$, which contained the measured exchange fluxes in carbon-mole units. For consumed metabolites, the corresponding coefficients were negative. If all the metabolites were measured and a 100% elemental balance was given, these quantities would directly be stoichiometric coefficients of the macrochemical growth equation. Because of imperfect measurements, such a directly determined macrochemical growth equation would not be mass balanced. We therefore fit a mass-balanced macrochemical growth equation to the data in the following way:
From the experimentally measured catabolic products, we derived a list of net catabolic reactions, which resulted in the production of the observed metabolite. For example, if acetate was measured for growth on glucose, the assumed net catabolic reaction was$$C_{6}H_{12}O_{6}+2O_{2}\to 2C_{2}H_{4}O_{2}+2{\mathrm{CO}}_{2}+2H_{2}O,$$
which was further normalized to a carbon-mole as$${\left[{\mathrm{CH}}_{2}O\right]}_{\mathrm{glc}}+\frac{1}{3}O_{2}\to \frac{2}{3}{\left[{\mathrm{CH}}_{2}O\right]}_{\mathrm{ace}}+\frac{1}{3}{\mathrm{CO}}_{2}+\frac{1}{3}H_{2}O$$
and converted into a vector *v* that contained the stoichiometric coefficients. The coefficients of this vector were ordered so that the first n+1 entries corresponded to the experimentally measured metabolites, including the biomass. If *k* catabolic products were determined, this resulted in *k* vectors $v_{1}\ldots v_{k}$.An additional vector $v_{0}$ was defined by characterizing the biomass growth as follows: From the elemental composition of the biomass X, the degree of reduction $\gamma_{X}$ was determined using [45]. The degree of reduction of the carbon source S is denoted by $\gamma_{S}$. The idealized anabolic equation was then assumed to be(1)$$\begin{array}{cc} \mathrm{If}\;\gamma_{X}>\gamma_{S}:\; & \frac{\gamma_{X}}{\gamma_{S}}S\to X+\left(\frac{\gamma_{X}}{\gamma_{S}}-1\right){\mathrm{CO}}_{2} \end{array}$$(2)$$\begin{array}{cc} \mathrm{If}\;\gamma_{X}<\gamma_{S}:\; & S+\frac{1}{4}\left(1-\frac{\gamma_{X}}{\gamma_{S}}\right)O_{2}\to X. \end{array}$$
If the biomass was more reduced than the carbon source, we assumed that the electrons required for the reduction were obtained by the oxidation of some nutrient carbons to CO_2_. If the biomass was more oxidized, we assumed that oxygen was available as an electron acceptor. This assumption excludes anaerobic growth conditions in which the biomass is more oxidized than the carbon source. We did not include such cases in our present analysis. It has to be stressed that these idealized anabolic equations are not fully mass balanced because they do not consider the nitrogen, phosphorus, or sulfur balances. However, they are fully carbon balanced and the calculation of the degree of reduction using Roels [45] considers the reduction state of the nitrogen source.A linear combination was determined that fit the experimental vector *e* best by minimizing the residual squares:(3)$$\min_{\alpha_{i}}\sum_{j=0}^{n}{\left(e_{j}-\sum_{i=0}^{k}\alpha_{i}v_{i,j}\right)}^{2},\;\mathrm{such}\;\mathrm{that}\;\alpha_{i}\geq 0.$$
These residuals are reported, e.g., in Table 1. The fitting procedure was performed in Python (v. 3.11.4) with the method scipy.optimize.minimize.The resulting vector *m*, which when normalized to one unit of biomass formed, contains the coefficients(4)$$m_{j}=\frac{\sum_{i=0}^{k}\alpha_{i}v_{i,j}}{\sum_{i=0}^{k}\alpha_{i}v_{i,0}}.$$
These define the macrochemical growth equation (per C-mol of biomass) that best fits the experimentally measured exchange fluxes. The combination of catabolic routes (represented by the vector $v_{\mathrm{NCR}}=\sum_{i=1}^{k}\alpha_{i}v_{i}$) forms the NCR, which is used for further calculations.

The assumed catabolic reactions, as well as the idealized anabolic reactions, are theoretical abstractions, without guarantee that these routes can actually be realized. However, the definitions provided here were tailored for the purpose of this study to compare the catabolic energy requirement to form biomass across species and conditions. It is not trivial to separate anabolism and catabolism, especially considering the multitude of amphibolic reactions [46]. We therefore decided to provide a clear mathematical definition based on the fact that cellular metabolism can be considered as a linear combination of elementary pathways [47]; we subsequently determined a linear combination that fit the experimental data best. The catabolic routes, as well as the idealized anabolic reactions, thus form a convex basis of the search space of all permissible fluxes. Therefore, the key conditions are that the definitions of the routes are unique, where the routes are carbon- and redox-balanced and linearly independent. We specifically decided to include the redox burden in the anabolic route (see Equation (Equation 1)) because this definition is simple, unique, and results in a straight-forward redox-balanced chemical equation. In the case that biomass is more oxidized than the carbon source, we assumed that oxygen is available as an electron acceptor (see Equation (2)). This assumption limits the applicability of our theory, excluding anaerobic growth on reduced carbon sources. The publications we have identified as data sources do not include such cases.

### 4.2. Net Catabolic Gibbs Free Energy $\Delta G_{X/S}$

The idealized anabolic Equation (Equation 1) or (Equation 2) is subtracted from the vector *m*, yielding the vector $v_{\mathrm{NCR}}$ describing the NCR. The vector $v_{\mathrm{NCR}}$ represents a fully mass-balanced chemical conversion that is normalized to the formation of one carbon-mole of biomass. The corresponding energy of reaction is exactly the net catabolic Gibbs free energy $\Delta G_{X/S}$ required to produce one carbon-mole of biomass. Because the NCR is determined from the experimentally measured macrochemical equation, it is an empirical quantity, which comprises non-anabolic processes in metabolism and also includes maintenance processes and futile cycles. This value is determined using the eQuilibrator API [9], taking environmental data, such as pH and temperature, into account, as reported in the original literature. If not otherwise stated, a standard Mg^2+^ concentration of 1 mmol L^−1^ and an ionic strength of 0.25 mol L^−1^ were assumed. Because in most experiments, extracellular concentrations of nutrients and catabolic products have not been measured, we determined the reaction energies for biochemical standard conditions with concentrations at 1 mmol L^−1^. While the actual concentrations will certainly affect the reaction energies quantitatively, this effect can be expected to be rather small. As argued in [30], every order of magnitude deviation from the biochemical standard concentration will result in a shift of ∼5.7 kJ mol^−1^. With catabolic reaction energies of 300 kJ mol^−1^ and, in some cases, considerably more, even an uncertainty in concentrations of several magnitudes will result in only a rather small relative error. While it would be interesting to systematically measure concentrations, and thus, quantify the concentration-dependent effect, including measured concentrations would not qualitatively alter our results, and therefore, also not affect the conclusions we draw here.

### 4.3. Estimation of Errors

The group contribution method underlying the estimation of reaction energies [9,48,49] naturally involves significant uncertainties. However, these uncertainties are systematic and effect all datasets similarly. Because we were interested in comparing catabolic free energies across experiments, organisms, and conditions, we decided to ignore this systematic error but rather estimated the error that stemmed from the measurement uncertainties, which are usually indicated by a carbon recovery coefficient deviating from 100%. We estimated the error as follows: We compared the fitted substrate uptake rate (in vector *m*) with the experimentally measured rate (in vector *e*) and made two extreme assumptions to ‘correct’ this difference. For simplicity, we assumed the fitted uptake rate was smaller than the experimental rate (the argument below works exactly the same, but the opposite, if the rate is larger). First, we assumed that the missing carbon was completely converted into biomass, added this conversion to the vector *m*, and repeated the subsequent calculation. This resulted in a lower value of $\Delta G_{X/S}$ because the catabolic free energy was now normalized to a higher biomass production rate. Second, we assumed that the missing carbon was completely respired to CO_2_ and added the corresponding respiration reaction to the vector *m* before performing the subsequent calculations again. This resulted in a higher value of $\Delta G_{X/S}$ because the additionally respired carbon was added to the catabolic reaction energy. These errors are given as error bars in Figure 2 and, e.g., reported in Table 2.

For the data on iron limitation [38], the gas exchange was not measured but instead inferred. Therefore, insufficient information is available for the above described error estimation procedure. We decided to omit the error bars for these data.

## 5. Conclusions

In this study, we have shown that a thermodynamics analysis of the NCR provides a powerful framework for understanding the energetic strategies of microbial metabolism. The assumptions made to derive the NCR are generic, and thus, applicable to a wide range of carbon sources and organisms. Our approach employs a number of approximations, such as assuming generic values for extracellular nutrient concentrations, ionic strength, and intracellular Mg^2+^ concentrations. In principle, if these values were experimentally quantified, these approximations would not be necessary. However, the error introduced by these assumptions is small and will not qualitatively change our observations, and therefore, our conclusions. Examining the Gibbs free energy changes in the NCR across diverse microbes and nutrient limitations shows that microbes consistently use more free energy to produce biomass under anabolic limitations than under carbon limited conditions. This thermodynamic shift is not merely a passive consequence of nutrient limitation but we hypothesize that this is an active cellular strategy to optimize the proteome resource allocation and uptake rate, or the apparent $K_{M}$ of the cell for the limiting nutrient.

Our findings extend existing theories on the thermodynamic interpretation of carbon-limited growth at various growth rates to other nutrient limitations, such as nitrogen, sulfate, potassium, and phosphate. The observations suggest that thermodynamic optimization of the catabolic pathways used under such anabolic nutrient limitations is a general principle shaping microbial physiology. These insights pave the way for a broader understanding of how thermodynamic and proteomic constraints co-determine microbial fitness in specific nutrient-limited environments. Future work combining experimental flux measurements, proteomic profiling, and thermodynamic modeling under defined nutrient limitations will be crucial to validate and further refine these hypotheses. Ultimately, integrating thermodynamics into systems biology not only deepens our comprehension of microbial adaptation but also holds promise for the rational design of microbial strains for biotechnology and bioprocessing applications.

Understanding microbial thermodynamics has profound implications for bioproduction and industrial microbiology. A quantitative understanding how energetic requirements for growth change as a result of nutrient limitations will allow for predicting the energy dissipation, and therefore, the heat production, given the imposed nutrient limitations. Thermodynamic constraints are encountered in a large range of biotechnological applications. In vinegar production, acetic acid bacteria operate under oxygen-rich conditions with high metabolic fluxes, generating substantial heat that must be dissipated to maintain process stability. Precision fermentation, which is used for producing high-value compounds such as enzymes, flavors, and pharmaceuticals, often involves tightly regulated nutrient limitations to steer metabolic fluxes toward desired products. These conditions can lead to elevated catabolic energy release and increased thermal output. Moreover, flavor molecule biosynthesis, such as esters or terpenoids, or omega-3 production often require nutrient stress or specific metabolic states that enhance product yield but may also intensify heat generation. The concept of the NCR offers a theoretical foundation with which these thermodynamic shifts can be understood and the associated heat production can be predicted. Such an understanding will be crucial for designing effective cooling strategies and ensuring scalability of these industrial processes.

Expanding the current framework beyond hydrocarbon-based catabolism to include alternative electron donors, such as ammonia, would further enhance its applicability. Extending the theory to ammonia oxidation is conceptually straightforward, as the separation of the macrochemical growth equation into anabolic and catabolic components remains valid. However, this would require the experimental determination of exchange rates for nitrogen-containing metabolites. Future research should explore the integration of thermodynamic modeling with synthetic biology approaches, particularly in the context of precision fermentation and metabolic engineering.

## Figures and Tables

**Figure 1 ijms-26-10921-f001:**
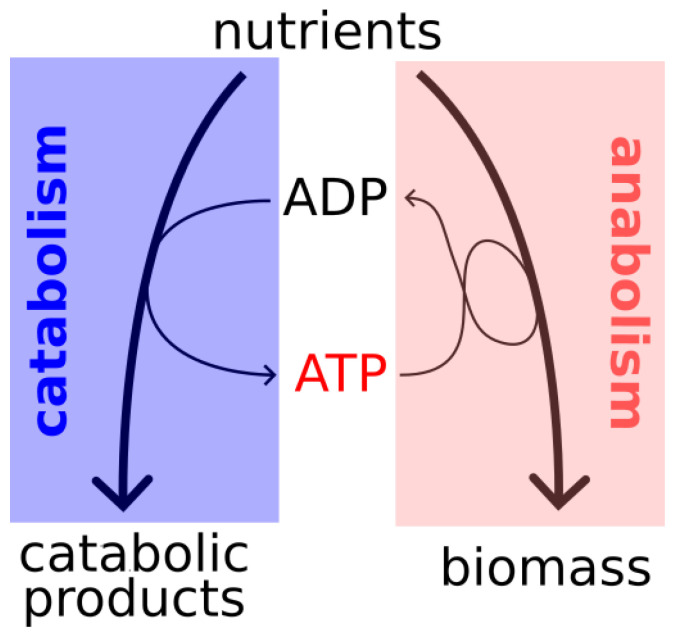
Microbial growth via energy conversion.

**Figure 2 ijms-26-10921-f002:**
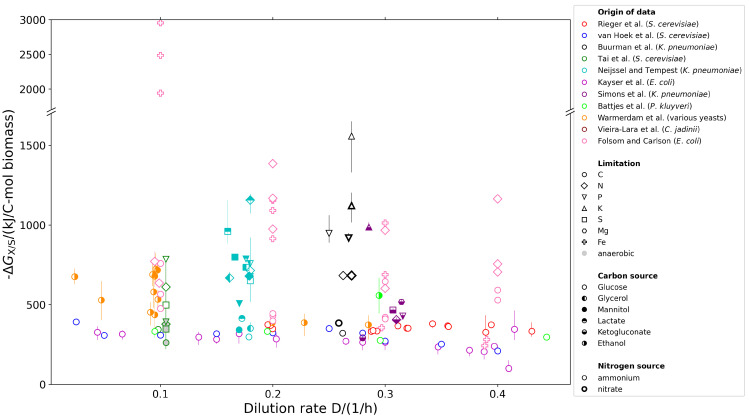
The calculated Gibbs free energy ($\Delta G_{X/S}$) of the NCR during various growth conditions. The error bars were estimated as described in the Methods by correcting mismatching carbon recovery rates by either using the ideal anabolic reaction or pure respiration. The data was retrieved from the following original publications: Rieger et al. [18], van Hoek et al. [19], Buurman et al. [12], Tai et al. [25], Neijssel and Tempest [27], Kayser et al. [35], Simons et al. [31], Battjes et al. [34], Warmerdam et al. [33], Vieira-Lara et al. [32], Folsom and Carlson [38]. Data from Folsom and Carlson [38] did not include gas exchange rates; therefore, the carbon recovery coefficient was not known, and thus, this approach was not possible.

**Figure 3 ijms-26-10921-f003:**
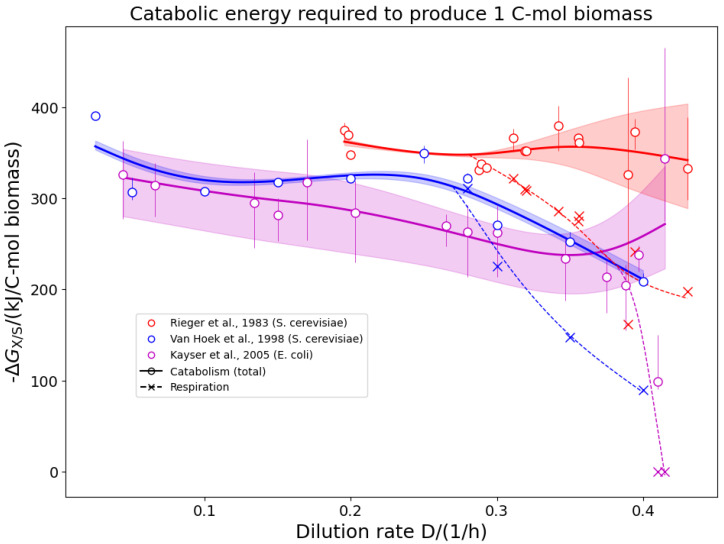
Calculated Gibbs free energy change ($\Delta G_{X/S}$) of the NCR under carbon-limited conditions, showing both the total free energy for catabolism and the contribution from the respiratory component. The data has been obtained for three classical overflow metabolism experiments. In Rieger et al. [18] and van Hoek et al. [19] the yeast *S. cerevisiae* is grown under different dilution rates. Ethanol production is observed for dilution rates ≳ 0.3 h^−1^. In Kayser et al. [35], *E. coli* was grown under different dilution rates, and acetate production is observed for growth rates ≳ 0.4 h^−1^.

**Table 1 ijms-26-10921-t001:** Summary of rates and thermodynamic properties under different continuous growth conditions and nutrient limitations of *Saccharmoyces cerevisiae* (data extracted from [25]). The table gives a schematic overview of respiration and fermentation rates (expressed as C-mol glucose/C-mol biomass/h) and the calculated overall thermodynamics of the net catabolic reaction (NCR) for each nutrient-limited condition. The quality of fit reports the sum of the residual squares of the fitted versus experimental fluxes (measured in $C-\mathrm{mol}\;L^{-1}\;h^{-1})$. Higher numbers indicate a carbon recovery coefficient that diverges from 100% and, as a result, a poorer fit to the data.

Condition	Limitation	Respiration	Fermentation	Δ*G* (kJ/C–mol Biomass)	Quality of Fit ($r^{2}$)
Aerobic	Carbon	0.7	0	−344	0.02
	Nitrogen	0.79	6.17	−610	3.03
	Phosphorus	1.15	6.17	−784	5.22
	Sulfur	0.77	3.41	−497	1.23
Anaerobic	Carbon	0	7.27	−260	2.22
	Nitrogen	0	10.50	−376	8.73
	Phosphorus	0	10.50	−391	10.70
	Sulfur	0	9.65	−346	15.59

**Table 2 ijms-26-10921-t002:** Summary of fermentation thermodynamics under different continuous growth conditions and nutrient limitations of *Klebsiella pneumoniae* [12,31]. The table provides a schematic overview of the calculated overall thermodynamics of the growth reaction as described above across various nutrient-limited continuous growth conditions. The errors were estimated by correcting the discrepancy between the fitted and measured fluxes either with biomass production fluxes or with respiration fluxes only.

Condition	Limitation	Δ*G* (kJ/C–mol Biomass)
Ammonia	Carbon	−321±14
	Nitrogen	−682±1
	Phosphorus	−946±87
	Potassium	−1557±160
Nitrate	Carbon	−384±1
	Nitrogen	−682±15
	Phosphorus	−918±29
	Potassium	−1121±93

**Table 3 ijms-26-10921-t003:** The driving force for ATP formation during heterotrophic growth for different catabolic pathways. The calculation of the ΔG was performed using eQuilibrator [9]. Values are given for biochemical standard conditions assuming a concentration of 1 mmol L^−1^.

Organism	Reaction	$\Delta_{r}$ $G^{1\mathrm{mM}}$ (kJ/mol ATP)	Growth
*L. lactis*	Glucose + 3 ADP + 3 P_i_ → Acetate + Ethanol + 2 Formate + 3 ATP + 2 H_2_O	−43.8	Slow
*L. lactis*	Glucose + 2 ADP + 2 P_i_ → 2Lactate + 2 ATP + 2 H_2_O	−59.4	Fast
*S. cerevisiae*	Glucose + 36 ADP + 36 P_i_ + 6 O_2_ → 6 CO_2_ + 36 ATP + 36 H_2_O	−38.2	Slow
*S. cerevisiae*	Glucose + 2 ADP + 2 P_i_ → 2 Ethanol + 2 CO_2_ + 2 ATP + 2 H_2_O	−93.9	Fast
*E. coli*	Glucose + 28 ADP + 28 P_i_ + 6 O_2_ → 6 CO_2_ + 28 ATP + 28 H_2_O	−61.6	Slow
*E. coli*	Glucose + 5 ADP + 5 P_i_ + 2 O_2_ → 2 Acetate + 2 CO_2_ + 5 ATP + 5 H_2_O	−198.7	Fast

## Data Availability

All code developed for this project is found at https://gitlab.com/qtb-hhu/thermodynamics-task-force/2025-catabolic-deltag-for-various-limitations (accessed on 5 November 2025). Files and instructions are given showing how to reproduce the results and figures in this publication.

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
