# Peer review of "Integrative Thermodynamic Strategies in Microbial Metabolism"

_ijms, 2025, doi:10.3390/ijms262210921_

Round 1
Reviewer 1 Report
Comments and Suggestions for Authors
Dear Authors,
Please find below my recommendations for "Integrative Thermodynamics Strategies in Microbial Metabolism" manuscript.
- L32-34: In my opinion this statement “optimization toward maximum growth rate rather than yield.” is overgeneralized as this is condition-dependent. Organisms often trade off rate versus yield, but the generality varies with nutrient limitation, stress, and ecological context. In my opinion presenting it as a universal conclusion risks overstating the claim unless scope-limited
- L34: YATP discussion is historically specific although contemporary bioenergetics often uses maintenance coefficients (mATP) and P/O variability in systems biology contexts. The text implies deviations are primarily maintenance and futile cycling; however, authors could consider that ion leakage, redox balancing, and variable coupling efficiencies also contribute
- L38-52: In my opinion the ED vs EMP rationale is simplified by producing only one instead of two ATP, ΔG is more negative, allowing higher forward kcat values. Thermodynamic driving force and enzyme saturation are influenced by metabolite activities and network context. Up to my knowledge the link between pathway ATP yield and uniformly more negative pathway ΔG is not guaranteed across physiological metabolite ranges. I think this needs nuance or a conditional statement
- “Idealized anabolism” assumes maximal theoretical carbon yield and external electron acceptors when biomass is less reduced than substrate. In my opinion this construction may bias ΔG_X/S by effectively reassigning part of the redox burden to catabolism irrespective of actual physiology. In anaerobic or electron-acceptor-limited conditions, the assumption breaks down and the NCR will be condition-specific. I recommend that the Introduction delineate applicability (aerobic vs anaerobic; respiratory vs fermentative regimes) if possible
- L73: Separation of anabolism by an “ideal” anabolic reaction: Subtracting an idealized anabolic step from the macro reaction to define NCR excludes the energetic costs of biosynthesis (ATP, NAD(P)H, ion gradients) from the ΔG attribution. In my opinion, this risks internal inconsistency: catabolism must supply that energy; removing it may bias ΔG_X/S toward more negative values and conflate “catabolic” driving force with a bookkeeping artifact. For me, the framework should explicitly reconcile energy and redox cofactor demands of anabolism with the NCR definition, or provide a proof that ΔG_X/S so defined preserves the true net catabolic work per biomass formed
- L102-104 (eQuilibrator): Standard transformed Gibbs energies depend on pH, ionic strength, temperature, Mg2+ binding, and reactant activities. The Theory does not specify the default conditions nor how growth media conditions are mapped. Because ΔG is log-linear in activities, ignoring realistic ranges can shift ΔG_X/S by tens of kJ/C-mol. A clear statement of assumed conditions and a sensitivity analysis outline are needed in my opinion
- L117: First step assumes specific net catabolic reactions (e.g., glucose + O₂ → acetate + CO₂ + H₂O) without justification. These stoichiometries may not reflect actual metabolic pathways; for instance, acetate production from glucose can occur via multiple routes (overflow metabolism, mixed-acid fermentation) with different O₂ dependencies and cofactor requirements. In my opinion the method should either derive these from pathway analysis or acknowledge the approximation
- I am not sure about the idealized anabolic equation therefore I invite the authors to verify. Eq 1. If biomass is more reduced than substrate, additional reducing equivalents are needed, not CO₂ production if; The factor (1/4) assumes 4 electrons per O₂, but the stoichiometry doesn't account for water formation or proton balance. For γX > γS, the reaction should consume external electron donors or produce oxidized byproducts; for γX < γS, excess reducing equivalents must be oxidized
- L382-383: I my opinion this statement should be reconsidered as the NCR derivation in this manuscript: omits explicit N, P, S assimilation energetics and their redox/ATP requirements, which vary widely across organisms and conditions (e.g., ammonium vs nitrate vs N2; sulfate vs cysteine; phosphate uptake vs polyphosphate metabolism); uses a fixed or inadequately specified biomass elemental composition γX despite known dependence on growth rate and limitation regime; and treats activities and standard-state corrections uniformly, without specifying pH, ionic strength, Mg2+ binding, or temperature for each dataset. These are not universal assumptions; they are approximations that can bias ΔG estimates in condition- and organism-specific ways in my opinion
- In my opinion a subsection dedicated for limitations could be considered by authors
Reviewer 2 Report
Comments and Suggestions for Authors
The manuscript presents a valuable thermodynamic perspective on microbial metabolism under different nutrient limitations and introduces the Net Catabolic Reaction (NCR) concept as a unifying descriptor for comparing Gibbs free energy changes across diverse growth conditions. The study is timely and relevant to systems biology and microbial bioenergetics. However, substantial revisions are needed to enhance its scientific rigor, clarity, and interpretability before it can be considered for publication.
- The manuscript lacks a clear quantitative demonstration of novelty. While the authors apply thermodynamic analysis to non-carbon limitations, the theoretical framework remains largely derivative of prior works on microbial energy balance and proteome allocation. A precise mathematical or conceptual innovation beyond the established Gibbs free energy–based models must be identified and justified.
- The NCR concept, though promising, is insufficiently validated. The authors assume that the net catabolic reaction accurately reflects overall energetic fluxes, yet this abstraction ignores intracellular heterogeneity and metabolic network redundancy. It would strengthen the paper to demonstrate, using model verification or sensitivity analysis, that NCR-derived ΔG_X/S values correlate with experimentally measured energy yields or ATP maintenance coefficients.
- Dataset integration is problematic due to inconsistent experimental parameters among sources. Combining data from different organisms, dilution rates, and growth conditions without rigorous normalization introduces systematic bias. The authors should implement a standardized normalization pipeline, possibly using dimensionless thermodynamic efficiency ratios or scaling factors to correct inter-study variability.
- The treatment of biochemical standard conditions (1 mM metabolite concentrations) oversimplifies physiological reality. Because intracellular metabolite levels vary between 10⁻⁴ and 10⁻² M, the computed ΔG_X/S values likely deviate by tens of kJ·mol⁻¹. The authors should perform a quantitative error propagation or Monte Carlo simulation to estimate the magnitude of uncertainty introduced by this assumption.
- Maintenance energy and futile cycling are central to thermodynamic modeling but are not explicitly incorporated in the NCR approach. The model currently assumes steady-state efficiency without accounting for maintenance ATP flux (m_ATP) or dissipative reactions, which could significantly alter ΔG_X/S estimations. Incorporating a term for non-growth-associated energy demand is essential for biological realism.
- The interpretation of more negative ΔG_X/S under nutrient limitations is largely qualitative. The three proposed hypotheses (proteome allocation, coupled transport, bioenergetic efficiency) are plausible but require quantitative linkage to known physiological metrics—such as ATP/ADP ratios, membrane potential (Δψ), or transporter stoichiometry—to confirm that observed thermodynamic shifts reflect real cellular adaptations.
- The use of literature datasets without independent experimental validation limits the reliability of the findings. Ideally, at least one representative system (e.g., coli or S. cerevisiae) should be recalculated using experimentally measured exchange fluxes to verify that the predicted ΔG_X/S aligns with measured respiratory quotients or carbon recoveries.
- Statistical rigor is insufficient. The manuscript lacks confidence intervals, variance estimates, or hypothesis testing to confirm that the observed differences in ΔG_X/S among nutrient conditions are statistically significant. Error propagation through each computational step (stoichiometric fitting → NCR derivation → ΔG calculation) should be quantitatively reported.
- Figures 2 and 3 inadequately represent the dataset. Axis units, sample sizes, organism labels, and limitation types must be clearly indicated. Data dispersion and error ranges should be visualized to distinguish biological variation from methodological noise. A regression or correlation analysis between ΔG_X/S and nutrient limitation strength would provide a more quantitative overview.
- The manuscript’s discussion of biotechnological relevance is speculative. The authors should avoid general claims about “process optimization” and instead demonstrate, via thermodynamic modeling or flux simulation, how manipulating ΔG_X/S could predict or improve energy yields in specific bioprocesses (e.g., nitrogen-limited yeast fermentations). This would elevate the work from theoretical interpretation to practical application.
Comments on the Quality of English Language
The English is generally clear but requires moderate editing to improve sentence flow, clarity, and grammatical precision.
Round 2
Reviewer 1 Report
Comments and Suggestions for Authors
Dear Authors,
Thank you for improving the "Integrative Thermodynamics Strategies in Microbial Metabolism" manuscript and for the provided clarifications.
Author Response
We thank the reviewer for their thorough and helpful comments.
Reviewer 2 Report
Comments and Suggestions for Authors
The revised manuscript shows clear improvement in presentation, organization, and scientific explanation. The concept of the Net Catabolic Reaction (NCR) is now better justified, and the responses to previous reviewer concerns are well addressed. The inclusion of standardized normalization procedures, clearer error estimation, and improved figures have strengthened the paper.
Minor points for final revision:
- The industrial relevance section could be slightly expanded with a simple, practical example connecting the NCR findings to bioprocess or fermentation optimization.
- Figure captions could be shortened and simplified for easier interpretation.
- Please ensure consistency of symbols and units throughout the manuscript (e.g., ΔG, NCR).
- A brief language polishing pass is recommended to improve fluency and reduce redundancy.
Overall, the manuscript is now scientifically sound, clear, and nearly ready for publication after addressing these minor editorial adjustments.
